# Forsythiaside A Activates AMP-Activated Protein Kinase and Regulates Oxidative Stress via Nrf2 Signaling

**DOI:** 10.3390/ijms242317033

**Published:** 2023-12-01

**Authors:** Young Eun Kim, Seon-Been Bak, Min-Jin Kim, Su-Jin Bae, Won-Yung Lee, Young Woo Kim

**Affiliations:** 1AI-Bio Convergence DDI Basic Research Lab., School of Korean Medicine, Dongguk University, Gyeongju 38066, Republic of Korea; kye9912@naver.com (Y.E.K.); sbpark@dongguk.ac.kr (S.-B.B.); alswls2055@naver.com (M.-J.K.); realsujin@naver.com (S.-J.B.); wonyung21@naver.com (W.-Y.L.); 2College of Korean Medicine, Wonkwang University, Iksan 54538, Republic of Korea

**Keywords:** AMPK, Nrf2, HO-1, forsythiaside A, liver injury

## Abstract

Forsythiaside A (FA) is an active constituent isolated from *Forsythia suspensa*, a beneficial herb used in traditional medicine known for its antioxidant and anti-inflammatory properties. Although various studies have suggested that FA has the protective effects, its impacts on arachidonic acid (AA) plus iron in vitro models and carbon tetrachloride (CCl₄)-induced mouse liver damage in vivo have not been explored. In this study, HepG2 cells were subjected to AA + iron treatment to induce apoptosis and mitochondrial impairment and determine the molecular mechanisms. FA exhibited protective effects by inhibiting cell damage and reactive oxygen species (ROS) production induced by AA + iron, as assessed via immunoblot and flow cytometry analyses. Further molecular investigations revealed that FA resulted in the activation of extracellular-signal-related protein kinase (ERK), which subsequently triggered the activation of AMP-activated protein kinase (AMPK), a critical regulator of cellular oxidative stress. Additionally, FA modulated the nuclear factor erythroid 2-related factor 2 (Nrf2) signaling pathway, which is a significant antioxidant transcription factor regulated by the AMPK pathway. For in vivo studies, mice were orally administered FA and then subjected to induction of CCl₄-based hepatotoxicity. The protective effect of FA was confirmed via blood biochemistry and immunohistochemical analyses. In conclusion, our findings demonstrated the protective effects of FA against oxidative stress both in vitro and in vivo, thus indicating that FA is a potential candidate for liver protection. Our study sheds light on the mechanistic pathways involved in the antioxidant effects of FA, highlighting the hepatoprotective potential of naturally occurring compounds in traditional herbs, such as FA.

## 1. Introduction

Oxidative stress is caused by an excessive production of reactive oxygen species (ROS) and reduced antioxidant production [1]. ROS are derived from exogenous as well as endogenous sources, including mitochondria, xanthine oxidase, cytochrome p450 metabolism, peroxisomes, and inflammatory cell activation [1]. Normal levels of ROS production regulate cell functions such as proliferation, defense against signaling pathways, chip-in microorganisms, and gene expression [2]. However, excessive ROS production is a major cause of chronic liver disease, which causes oxidative stress leading to lipid, protein, and DNA alteration, and it affects pathways that regulate biological function, thereby causing liver damage [3].

Oxidative stress due to the accumulation of iron causes various chronic liver diseases such as liver fibrosis, cirrhosis, and hepatocellular carcinoma [4]. Excessive iron accumulation catalyzes the release of arachidonic acid (AA) and eicosanoids, thereby triggering oxidative stress, inflammatory reactions, and cell death [5]. AA is abundant in the liver and increases the iron content, thereby leading to hepatocyte toxicity by increased oxidative stress and consequent mitochondrial damage [6,7]. Therefore, it is crucial to explore candidate substances that can protect against the hepatotoxicity caused by iron and AA.

Notably, there are key signaling pathways for the regulation of the cellular damage. Nuclear factor erythroid 2-related factor 2 (Nrf2) plays an important role in cellular and biological defenses against oxidative stress, such as detoxification, regulation of cell metabolism, and promotion of cell proliferation, which are important in the pathogenesis of various diseases [8]. In addition, Nrf2 contains seven NRF2–erythroid cell-derived protein with Cap ‘N’ Collar (CNC) homology (ECH) homolog (Neh1–7) domains [9]. Under normal conditions, the negative regulatory domain Neh2 binds to cytoplasmic kelch-like ECH-associated protein 1 (Keap1), leading to the dissociation of Nrf2 and Keap1 from the cell cytoplasm [10,11]. Keap1 is a negative regulator of Nrf2 and maintains low physiological activity of Nrf2 by mediating the proteasome-mediated decomposition of Nrf2 via ubiquitination [12]. Previous studies showed that AMP-activated protein kinase (AMPK) may be involved in the Nrf2 regulation for the induction of antioxidants detoxifying enzymes [13,14].

*Forsythia suspensa* (FS) exhibits various pharmacological activities, including anti-inflammatory, antioxidant, and liver protective effects [15]. Forsythiaside A (FA), an active ingredient isolated from FS, is a major indicator of physiological activity and exerts antioxidant, anti-inflammatory, antibacterial, antiviral, and nerve- and liver-protective effects [16,17,18,19,20,21]. Although there have been studies on the various functions of FA [22,23,24,25], there is a lack of research on the efficacy and underlying mechanism of the hepatoprotective effect of FA. Therefore, in this study, the antioxidant effects of FA on the damage caused by AA + iron and CCl_4_ induction, as well as underlying mechanisms related with AMPK and Nrf2, were examined.

## 2. Results

### 2.1. Effect of FA on AA + Iron-Induced Cytotoxicity

The effect of FA on AA + iron-induced cytotoxicity in HepG2 cells was assessed by an MTT assay. Cell viability was measured after treating HepG2 cells with varying levels of FA (1, 3, 10, and 30 μM). The cells treated with AA + iron exhibited reduced cell survival compared to the control. However, in cells treated with FA, cell viability increased with the concentration of FA (Figure 1A). Because cell viability was the highest at 30 μM FA, this concentration was used for further experiments. The protective effect of FA against AA + iron was confirmed using fluorescence microscopy. Live cells stained with calcein- acetomethoxy (AM) displayed red fluorescence, whereas dead cells stained with propidium iodide (PI) displayed green fluorescence (Figure 1B). The expression of PARP, procaspase-3, and other apoptosis-related protein markers was confirmed via Western blot analysis. Treatment with AA + iron reduced the expression of PARP and procaspase-3; however, this effect was blocked by FA treatment (Figure 1C). Intracellular ROS levels were measured using DCF-DA. ROS levels increased significantly in AA + iron-treated cells, and FA treatment reduced the generation of ROS induced by AA + iron. ROS production did not increase in cells treated with FA alone (Figure 1D).

### 2.2. Effect of FA on AA + Iron ROS Generation and Mitochondrial Dysfunction

The protective effect of FA against AA + iron-induced oxidative stress was also confirmed. Mitochondrial dysfunction induced by AA + iron treatment was verified via flow cytometry using rhodamine 123 (Rh 123), which specifically stains mitochondria. AA + iron treatment increased Rh 123-negative cells compared to the control, but FA pretreatment prevented this increase in Rh 123-negative cells by AA + iron exposure. There was no difference in the staining intensity between FA alone versus the control (Figure 2A), thus indicating that AA + iron protects hepatocytes against mitochondrial dysfunction. To assess the extent of cell death, HepG2 cells were stained with Annexin V/PI and analyzed via FACS. The results indicated an increase in cell death with AA + iron treatment compared to the control group. However, treatment with FA resulted in a decrease in cell death (Figure 2B). Furthermore, the mitochondrial membrane potential (MMP) was measured by staining HepG2 cells with JC-1. Treatment with AA + iron increased the ratio of green monomers, while treatment with FA decreased the ratio of green monomers (i.e., control, 45.97 ± 1.86%; AA + iron, 65.53 ± 4.85% **; AA + iron + FA, 51.97 ± 4.37% #, FA alone, 32.22 ± 5.83; the data represent the means ± S.D. of replicate experiments; the significance of the statistical differences between each treatment group and the vehicle-treated control group (** *p* < 0.01) or AA + iron-induced group (# *p* < 0.05) was verified) (Figure 2C). Additionally, mitochondrial superoxide levels were measured by staining HepG2 cells with MitoSOX Red. The results showed an increase (right shift) in mitochondrial superoxide production by AA + iron, which was reduced (left-shifted) by treatment with FA (Figure 2D).

To study the mechanism of FA protection related with mitochondrial ROS production, the effect of FA was examined in HepG2 cells in the absence of iron, because it has been shown that the cells undergo apoptosis in the presence of iron. Upon staining HepG2 cells with JC-1, it was observed that AA also increased the ratio of green monomers (i.e., control, 51%; AA, 63.9%), while treatment with FA decreased the ratio of green monomers (Figure 2E). Additionally, staining HepG2 cells with MitoSOX Red revealed that AA led to an increased production of mitochondrial superoxide compared to the control, whereas FA reduced the elevated mitochondrial superoxide induced by AA (Figure 2F). Therefore, FA decreased the increased ROS in both AA and AA + iron-induced experiments in HepG2 cells.

### 2.3. Effect of FA on AMPK Activation

Western blot was performed to confirm the mechanism by which FA affects AMPK activity in HepG2 and Huh-7 cells. Treatment of HepG2 cells with 30 μM FA-induced phosphorylation of AMPK in a time-dependent manner, with increased AMPK phosphorylation detected after 30 min of FA treatment, and acetyl-CoA carboxylase (ACC) phosphorylation, which is the major downstream target of AMPK occurring within 10 min to 1 h (Figure 3A). Similarly, AMPK and ACC in Huh-7 cells were also phosphorylated in a time-dependent manner by treatment with 30 μM FA (Figure 3B). In addition, LKB1, an enzyme upstream of AMPK, became phosphorylated after FA treatment (Figure 3C). Additionally, the MTT assay was performed to confirm whether FA protected LKB1-deficient HeLa cells from AA + iron-induced apoptosis. We observed that FA did not protect HeLa cells induced by AA + iron (Figure 3D), suggesting that FA protects cells from AA + iron-induced apoptosis through the activation of the AMPK pathway.

### 2.4. Effect of FA on Nuclear Nrf2 Activation

The transcription factor Nrf2 is important for cellular and biological defense mechanisms against oxidative stress [26]. We performed Western blotting of components of the Nrf2 signaling pathway to confirm the antioxidant mechanism of FA against oxidative stress. After FA treatment, the expression of Nrf2 increased in a time- and dose-dependent manner (Figure 4A,B). Next, we confirmed the effect of FA on the expression of Keap1, which inhibits Nrf2 activation. FA affected the expression of Keap1 in a time-dependent manner (Figure 4C). Activated Nrf2 binds to antioxidant response element (ARE) to activate the heme oxygenase (HO)-1 and protect cells from oxidative stress, and we found that FA increased HO-1 levels in a time-dependent manner (Figure 4D). Therefore, the effect of FA on the mRNA levels of HO-1 was examined. The results revealed that FA upregulated the mRNA expression of HO-1 compared to the control group (Figure 4E). Cells were treated with an inhibitor (Compound C) of AMPK, an upstream phosphorylation enzyme of HO-1, and suppression of HO-1 phosphorylation by FA treatment was confirmed via Western blotting (Figure 4F). Additionally, FA clearly induced HO-1 expression in HepG2 cells; however, the level of HO-1 was lower in LKB1-deficient HeLa compared to the level in HepG2 cells (Figure 4G). In conclusion, FA could activate Nrf2 to induce HO-1 expression, thereby suggesting that the Nrf2/HO-1 pathway is related to the AMPK signaling pathway.

### 2.5. Effect of FA on ERK Activation

Western blotting was performed to confirm the effects of FA on Akt and ERK, which are the upstream signaling molecules of AMPK. Treatment of the cells with FA upregulated the expression of Akt and ERK in a time-dependent manner (Figure 5A). Based on Western blotting, the treatment of cells with an inhibitor (PD98959) of ERK, the upstream phosphorylation enzyme of Nrf2, indicated that FA inhibited Nrf2 phosphorylation (Figure 5B). To confirm the effect of FA on ERK phosphorylation in AA + iron-induced mitochondrial dysfunction, cells were pretreated with the ERK inhibitor PD98059. The protective effect of FA against AA + iron-induced mitochondrial dysfunction was reversed by the ERK inhibitor (Figure 5C). These results suggest that ERK phosphorylation is partially associated with Nrf2 activity. To confirm the effect of FA on AA + iron-induced oxidative stress, an MTT assay was performed using isoliquiritigenin (ILQ) and sulforaphane (SFN), which are well-known for their antioxidant effects, as positive controls. FA, ILQ, and SFN increased cell viability, which was reduced by AA + iron treatment (Figure 5D). Therefore, overall, the results suggest that the antioxidative effect of FA in cells under conditions of oxidative stress induces HO-1 expression through the activation of Nrf2 via the ERK–AMPK signaling pathway (Figure 5E).

### 2.6. Effect of FA on Carbon Tetrachloride (CCl₄)-Induced Liver Injury

Next, we determined the effect of FA on liver injury induced by CCl_4_ in mice. CCl_4_ exposure is a representative model to screen hepato-protective reagents [13]. CCl_4_ injection markedly induced hepatocyte damage and inflammatory cell infiltration, as assessed via blood biochemistry and histology (Figure 6A,B). However, oral FA significantly inhibited the induction of liver injury by CCl_4_.

## 3. Discussion

Exposure to toxic substances and drugs can result in oxidative damage [5]. In this study, we used an acute liver injury model induced by iron and AA. Iron is the most abundant transition metal in the body, and it is essential for oxygen transport [27]. Iron is a component of numerous oxidase enzymes, and excessive accumulation of iron results in oxidative stress [28]. Moreover, it has been shown that iron + AA can induce severe damage in hepatocytes [29].

Mitochondria are involved in a variety of metabolic activities, including homeostasis, apoptosis activation, and apoptosis, and they are a major source of intracellular ROS [2,29,30]. Mitochondrial damage is caused by ROS produced by the mitochondria themselves, and the main enzyme components of the mitochondria are the tricarboxylic acid (TCA) circuit and respiratory or electron transport chains [31,32,33]. Oxidative damage occurs when enzymes such as glutathione peroxidase and peroxiredoxin III fail to convert ROS, such as superoxide radicals, into H_2_O [31,34]. The radicals produced in the TCA cycle can damage the mitochondria and reduce mitochondrial function by directly damaging mitochondrial proteins [31]. Damage to the mitochondria can increase cell demand for energy recovery processes, which can reduce mitochondrial function, and such mitochondrial damage can cause additional damage [31]. FA successfully prevented the damage induced by AA + iron in hepatocytes.

The cellular energy sensor AMPK is activated by various reactions that deplete cell energy levels, such as nutrient deficiency and exposure to toxins, which suppress hypoxic mitochondrial respiratory chain complexes and maintain cellular homeostasis in response to metabolic stress [35]. Additionally, AMPK is phosphorylated and activated at a specific threonine residue (Th-127), which is catalyzed by the upstream kinase LKB1 [36]. In response to oxidative stress, activated AMPK phosphorylates Nrf2, which is the main transcriptional regulator of the antioxidant gene program, thereby inducing nuclear accumulation of Nrf2 and the subsequent expression of associated antioxidant genes, thereby promoting cell survival [37,38].

Under conditions of oxidative stress or other chemical stimuli, phosphorylated Nrf2 is translocated to the nucleus and Keap1 dissociates to form a heterodimer, and Nrf2 then binds to the associated gene promoter ARE [39]. When Nrf2 is bound to ARE, it induces the expression of antioxidant enzyme genes, such as catalase, superoxide dismutase, glutathione-S-transferase NADH (nicotinamide adenine dinucleotide [NAD]+ hydrogen)–quinone oxidoreductase 1, thioredoxin reductase 1, and HO-1, which play roles in antioxidant stress, anti-inflammatory, anti-apoptotic, and cell protective responses [39,40,41]. HO-1 is a rate-limiting enzyme in the heme decomposition process, and HO-1 regulation by Nrf2 signaling plays an important role in the mechanism of oxidative damage, and it is activated by many stimuli related to oxidative stress [11].

Mitogen-activated protein kinases (MAPK) are activated as downstream effects of antioxidant reactions, and they in turn activate many transcription factors, such as Nrf2 [42,43]. Additionally, the MAPK pathway plays an important role in various cellular activities, including cell proliferation, differentiation, apoptosis, and survival [44,45]. The MAPK subfamily includes ERK, C-Jun N-terminal kinase, and p38 MAPK, which are members of the serine/threonine kinase group activated in response to various extracellular stimuli and mediate signaling from the cell surface to the nucleus [43,46]. The ERK pathway plays a pivotal role in internal metabolic stress, DNA damage, and cell signaling [44,47]. Generally, ERK is located in the cytoplasm and regulates gene expression by activating transcription factors, such as Nrf2 [44,48,49].

CCl_4_ is commonly used to study the effects of antioxidants on acute liver damage in animals [3,5,50]. CCl_4_ directly damages liver cells by changing the permeability of plasma, ribosome, and mitochondrial membranes, and it is oxidized by cytochrome P450 or cytochrome P450 2E1(CYP2E1) to form highly reactive free radical metabolites that cause radical stress [51,52,53], resulting in lipid peroxidation and liver damage, thereby leading to serious necrosis [51,54]. FA has been shown to affect the damage induced by CCl_4_ [23,55]. In the future, it will be necessary to clarify whether the protective effects of FA against CCl_4_ are mediated through AMPK and Nrf2.

Moreover, repeated liver injury can induce chronic liver injury. Excessive accumulation of collagen and extracellular matrix in the liver may lead to liver fibrosis [56]. Liver fibrosis is caused by chronic liver damage due to alcoholic fatty hepatitis, nonalcoholic fatty hepatitis, viral hepatitis, nonalcoholic fatty liver disease, and liver cirrhosis, and it can progress if not treated [3]. Additionally, liver fibrosis is caused by an imbalance between matrix synthesis and the decomposition of cells, which results in the accumulation of extracellular matrix deposition and fibrous scars that distort the liver structure [57,58]. Therefore, it is necessary to confirm the effects of FA on chronic liver diseases in future studies.

## 4. Materials and Methods

### 4.1. Reagents

Ferric nitrate, rhodamine 123 (Rh 123), 3-(4,5-dimethylthiazol-2-yl)-2,5-diphenyl-tetrazolium bromide (MTT), 2′,7′-dichlorofluorescein diacetate (DCFH-DA), Harris hematoxylin, and eosin were purchased from Sigma-Aldrich (St. Louis, MO, USA). AA was purchased from Calbiochem (San Diego, CA, USA). Anti-caspase-3, anti-full length PARP, anti-β-actin, anti-phospho-ACC, anti-phospho-AMPK, anti-phospho-liver kinase B1 (LKB1), anti-Nrf2, anti-phospho-ERK, anti-HO-1, anti-lamin A/C, and anti-phospho-ERK antibodies were obtained from Cell Signaling Technology (Danvers, MA, USA). Horseradish peroxidase-conjugated goat anti-rabbit and anti-mouse IgG antibodies were purchased from Enzo Life Sciences (East Farmingdale, New York, NY, USA).

### 4.2. Cell Culture

HepG2, Huh-7, and HeLa cells were obtained from the American Type Culture Collection (Rockville, MD, USA). HepG2 and HeLa cells were maintained in Dulbecco’s modified Eagle’s liquid medium with high glucose supplemented with 10% fetal bovine serum (FBS), 50 units/mL penicillin, and 50 μg/mL streptomycin. Huh-7 cells were maintained in Roswell Park Memorial Institute (RPMI) 1640 supplemented with 10% FBS, 50 units/mL penicillin, and 50 μg/mL streptomycin.

### 4.3. MTT Assay

HepG2 cells were plated in 48-well culture plates and incubated in FBS-free medium for 12 h. The cells were treated with FA (1–30 μM), AA (10 μM) for 12 h, and iron (5 μM) for 1 h [7]. After treatment, viable cells were stained with a 0.25 mg/mL MTT reagent for 1 h. The medium was removed, and the formed formazan crystals were dissolved in 200 μL of dimethyl sulfoxide. The absorbance was measured at 540 nm.

### 4.4. Measurement of ROS Production

HepG2 cells were treated with FA (30 μM), AA (10 μM) for 12 h, and iron (5 μM) for 1 h, and then incubated with 2′,7′-dichlorofluorescin-diacetate (DCF-DA) (10 μM) for 1 h. The cell fluorescence was measured as previously described [7].

### 4.5. Flow Cytometric Analysis

The MMP was measured via flow cytometry after staining with Rh 123. HepG2 cells were treated with FA (30 μM), AA (10 μM) for 12 h, and iron (5 μM) for 1 h, and then incubated with Rh 123 (0.05 μg/mL) for 1 h at 35 °C. The cells were trypsinized and centrifuged at 4 °C for 3 min at 3000 rpm. Cell MMP was detected using a BD Accuri™ C6 Plus flow cytometer (BD Biosciences, Franklin Lakes, NJ, USA) [5,7].

### 4.6. FACS Analysis of Annexin V and PI Staining

HepG2 cells were treated with FA (30 μM), AA (10 μM) for 12 h, and iron (5 μM) for 1 h. Then, the cells were trypsinized and harvested. HepG2 cell aggregates were washed with cold 1× PBS, followed by staining the cells with FITC Annexin V and PI at room temperature for 15 min. The extent of cell death was analyzed using the BD Accuri C6 Plus flow cytometer (BD Biosciences, Franklin Lakes, NJ, USA).

### 4.7. Mitochondrial Membrane Potential (ΔΨm) Analysis

The mitochondrial membrane potential (ΔΨm) was measured using JC-1. HepG2 cells were treated with FA (30 μM), AA (10 μM) for 12 h, and iron (5 μM) for 1 h. After treatment, cells were stained with JC-1 (10 μM) for 30 min at 35 °C. The stained cells were trypsinized and centrifuged at 3000 rpm for 3 min at 4 °C. The stained JC-1 cell aggregates were measured via flow cytometry [5,7].

### 4.8. Measurement of Superoxide in Mitochondria

To detect mitochondrial superoxide, MitoSox Red (Invitrogen, Rockford, IL, USA) was used. HepG2 cells were treated with FA (30 μM), AA (10 μM) for 12 h, and iron (5 μM) for 1 h. They were stained with MitoSox Red (1 μM) for 30 min at 35 °C in the dark. Fluorescence was detected via flow cytometry [5].

### 4.9. Preparation of Whole-Cell Lysate and Nuclear Extracts

Cells were lysed in a radioimmunoprecipitation assay (RIPA) buffer at 4 °C and then centrifuged at 15,000 rpm at 4 °C for 30 min to extract the cellular proteins, which were quantified using a bicinchoninic acid (BCA) protein assay kit. Nuclei were extracted using nuclear and cytoplasmic extraction reagents (NE-PER™) nuclear and cytoplasmic extraction reagents (Thermo Scientific, Rockford, IL, USA). The nuclear proteins were extracted using a nuclear extraction reagent (NER) buffer supplemented with protease inhibitors. The nuclear extracts were quantified using a BCA protein assay kit.

### 4.10. Western Blot Analysis

The quantified protein samples were electrophoresed using sodium dodecyl-sulfate-polyacrylamide gel electrophoresis and then transferred onto polyvinylidene fluoride membranes. The total protein amount was loaded at 30 μg. The primary antibodies were diluted at a ratio of 1:1000, and the secondary antibodies were diluted at a ratio of 1:5000. The bands were visualized using an enhanced chemiluminescence (ECL) reagent (Advensta, Menlo Park, CA, USA) and a ChemiDoc™ image analyzer (Vilber Lourmat, France).

### 4.11. Real-Time RT-PCR Analysis

The total RNA was isolated from HepG2 cells using TRIzol (Ambion, Life Technologies, Carlsbad, CA, USA). The RNA was reverse-transcribed to synthesize cDNA using a cDNA synthesis kit (Nanohelix, Daejeon, Republic of Korea). Real-time PCR was conducted using a qPCR kit (Nanohelix, Daejeon, Republic of Korea). The following primer sequences were used: human HO-1, 5′-CAGGCAGAGAATGCTGAGTTC-3′ (sense) and 5′-CATCACCAGCTTAAAGCCTT-3′ (antisense).

### 4.12. Animals and Treatment

All experimental animal procedures were approved by the Institutional Animal Care and Use Committee of Dongguk University (IACUC-2022-070-1). Male C57BL/6N mice (6 weeks old, 20–21 g) were purchased from Raon Bio (Yongin, Republic of Korea). The mice were randomly divided into five groups: vehicle-treated control, CCl_4_, CCl_4_ + FA 10 mg/kg, and CCl_4_ + FA 30 mg/kg. The mice were orally administered either FA (10 or 30 mg/kg, dissolved in 40% polyethylene glycol [PEG]) or vehicle (40% PEG) once daily for three days. A single intraperitoneal injection (0.5 mL/kg) [48,59] of CCl_4_ mixed with olive oil was administered 2 h after the final FA dose. After euthanasia with CO_2_, the liver and blood samples were prepared for the alanine transaminase (ALT) assay and histopathological analyses.

### 4.13. Statistical Analysis

The results are presented as the means ± the standard deviation and were subjected to analysis of variance (ANOVA) or *t*-tests to confirm statistical significance. The criterion for statistical significance was set at *p* < 0.05, *p* < 0.01, or *p* < 0.001.

## 5. Conclusions

FA reduced liver cell damage and mitochondrial damage induced by AA + iron. Furthermore, FA activated ERK, phosphorylating the AMPK signaling pathway, and phosphorylated AMPK activated Nrf2, thus activating antioxidant enzymes. These experiments demonstrate that the ERK-activated Nrf2 signaling pathway is the fundamental mechanism underlying FA’s antioxidant effects. Additionally, an oral administration of FA in mice inhibited liver damage caused by oxidative stress. Therefore, FA exhibits the potential as an antioxidant to protect against liver damage due to oxidative stress.

## Figures and Tables

**Figure 1 ijms-24-17033-f001:**
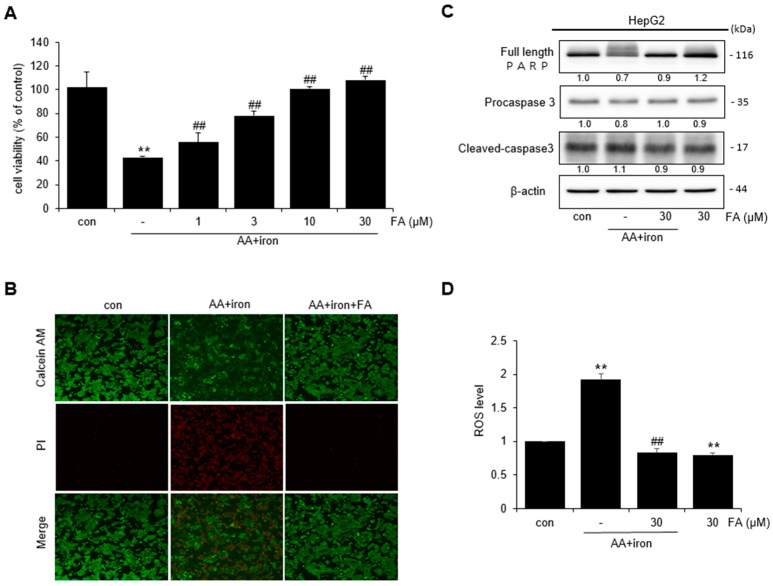
Effect of FA on AA + iron-induced cytotoxicity in HepG2 cells. (**A**) MTT assay: cell viability of HepG2 cells by FA (1–30 μM)-induced AA + iron. (**B**) Fluorescence image of HepG2 cells with calcein-AM/PI staining (×200). (**C**) Western blot analysis of apoptosis-related proteins. The HepG2 cells were treated with FA (30 μM), AA (10 μM) for 12 h, and iron (5 μM) for 1 h. (**D**) Measurement of ROS production via staining DCF-DA. HepG2 cells were treated with FA (30 μM), AA (10 μM) for 12 h, and iron (5 μM) for 1 h, and thereafter incubated with DCF-DA (10 μM) for 1 h. The data represent the means ± S.D. of replicate experiments. The significance of the statistical differences between each treatment group and the vehicle-treated control group (** *p* < 0.01) or AA + iron-induced group (## *p* < 0.01) was verified.

**Figure 2 ijms-24-17033-f002:**
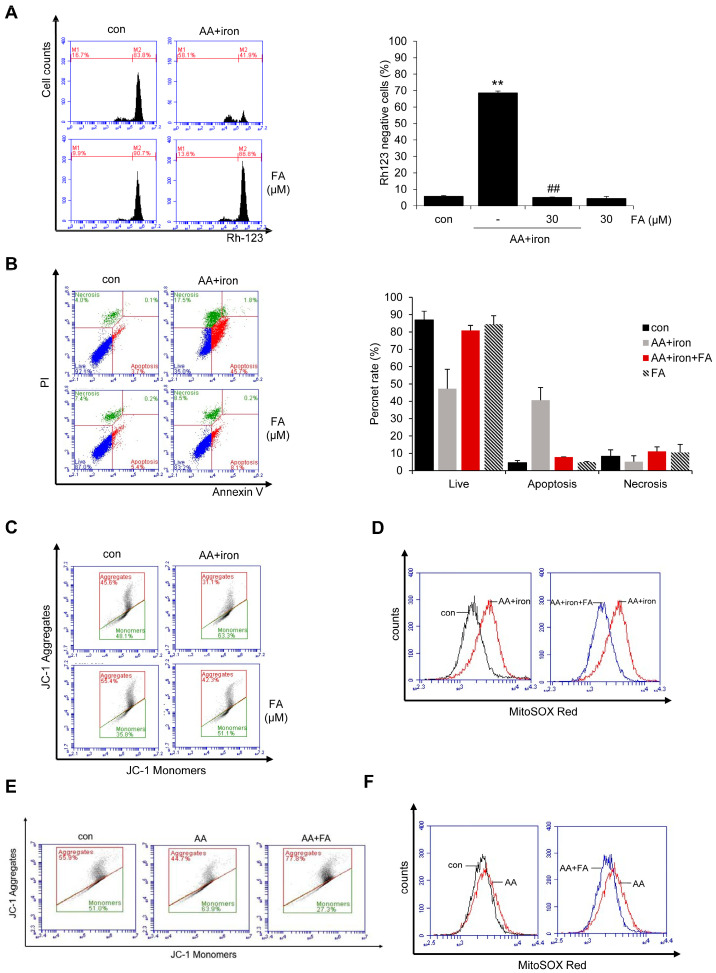
Effect of FA on AA + iron-induced ROS generation and mitochondrial dysfunction. (**A**) Flow cell analysis of Rh 123 staining. The HepG2 cells were treated as panel A, and thereafter incubated with Rh 123 (0.05 μg/mL) for 1 h. Relative cell population of low Rh123. (**B**) Flow cell analysis of Annexin V/PI staining. HepG2 cells were treated with FA (30 μM), AA (10 μM) for 12 h, and iron (5 μM) for 1 h. And cells were collected, followed by staining with Annexin V and PI at room temperature for 15 min. (**C**) Flow cell analysis of JC-1 staining. HepG2 cells were treated with FA (30 μM), AA (10 μM) for 12 h, and iron (5 μM) for 1 h. After staining with JC-1 (10 μM) for 30 min, the samples were analyzed via flow cytometry. (**D**) Flow cell analysis of MitoSOX red staining. HepG2 cells were treated with FA (30 μM), AA (10 μM) for 12 h, and iron (5 μM) for 1 h. After staining with MitoSOX red (1 μM) for 30 min, the samples were analyzed via flow cytometry. (**E**) Flow cytometry analysis for mitochondrial membrane potential. HepG2 cells were treated with FA (30 μM), AA (10 μM) for 12 h. (**F**) Flow cell analysis for mitochondrial superoxide. HepG2 cells were treated with FA (30 μM), AA (10 μM) for 12 h. The data represent the means ± S.D. of replicate experiments. The significance of the statistical differences between each treatment group and the vehicle-treated control group (** *p* < 0.01) or AA + iron-induced group (## *p* < 0.01) was verified.

**Figure 3 ijms-24-17033-f003:**
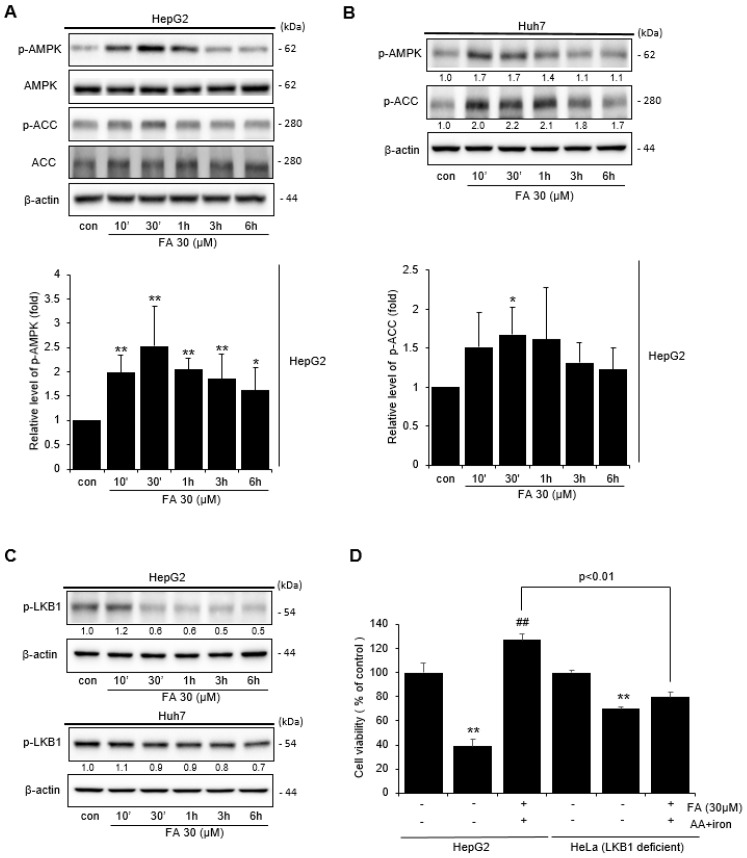
Activation of AMPK by FA. (**A**) Western blot analysis of phosphorylation of the AMPK signaling pathway by FA (30 μM). The HepG2 cells were incubated with FA (30 μM) for 10 min to 6 h. Relative band intensity of AMPK. (**B**) Huh-7 cells were treated with FA (30 μM) for 10 min to 6 h. (**C**) Western blot analysis of LKB1 phosphorylation by FA (30 μM). HepG2 and Huh-7 cells were treated with FA (30 μM) for 10 min to 6 h. (**D**) MTT assay of AA + iron-induced apoptosis of the HepG2 and LKB1-deficeint HeLa cells by FA (30 μM). The data represent the means ± S.D. of replicate experiments at least three times. The significance of the statistical differences between each treatment group and the vehicle-treated control group (* *p* < 0.05, ** *p* < 0.01) or AA + iron-induced group (## *p* < 0.01) was verified.

**Figure 4 ijms-24-17033-f004:**
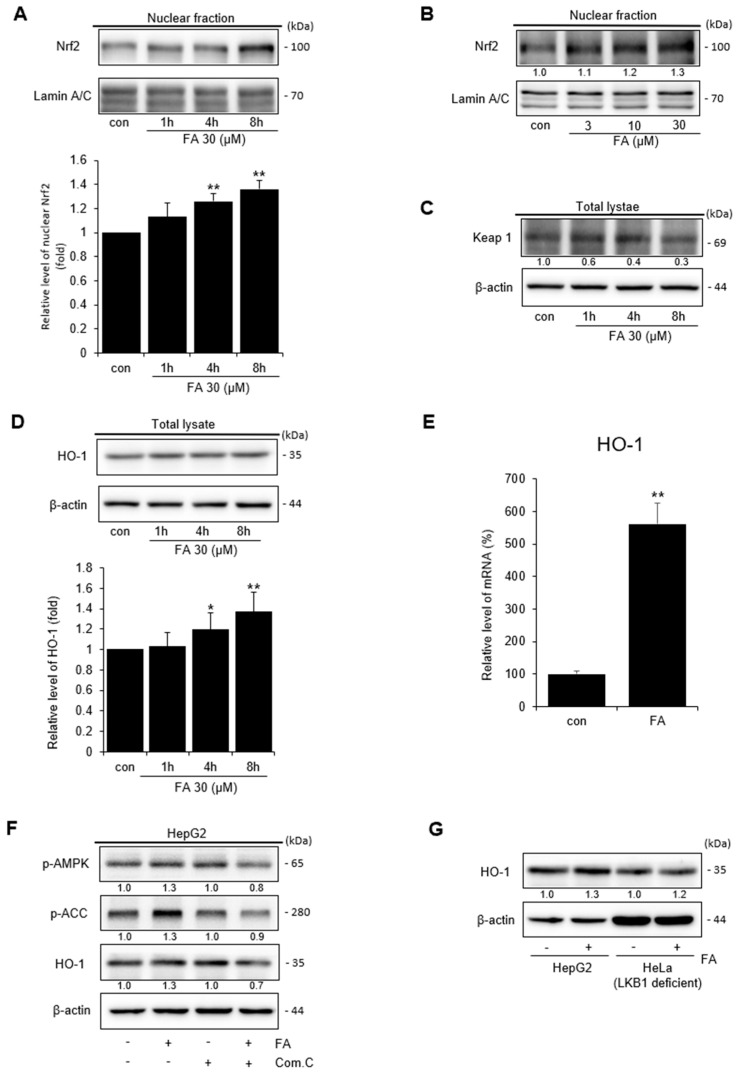
Nuclear Nrf2 activation by FA. (**A**) Western blot analysis of Nrf2 in the nucleus; Nrf2 was verified in the nuclear fraction of the cell treated with FA (30 μM) for 1–8 h. Western blotting for Lamin A/C also verified the same loading of protein. Relative band intensity of nuclear Nrf2. (**B**) Nrf2 was verified in the nuclear fraction of dose-dependent cells. The cells were treated with FA (3–30 μM) for 8 h. (**C**) Western blot analysis of Keap1 in the lysate cells. The HepG2 cells were treated with FA (30 μM) for 1–8 h. Western blot analysis of beta-actin verified the same loading of protein. (**D**) HO-1 was verified in the lysate cells. Cells were treated with FA (30 μM) for 1–8 h. Relative band intensity of HO-1. (**E**) Relative mRNA expression levels of the HO-1 target gene. HepG2 cells were treated with FA (30 μM) for 8 h. (**F**) Role of the AMPK pathway by FA to induce HO-1. HepG2 cells were pretreated with compound C for 1 h and thereafter treated with FA (30 μM) for 30 min. (**G**) Role of LKB1 by FA (30 μM) to induce HO-1 in HepG2 and LKB1-deficient HeLa cells. The data represent the means ± S.D. of replicate experiments at least three times. The significance of the statistical differences between each treatment group and the vehicle-treated control group (* *p* < 0.05, ** *p* < 0.01) was verified.

**Figure 5 ijms-24-17033-f005:**
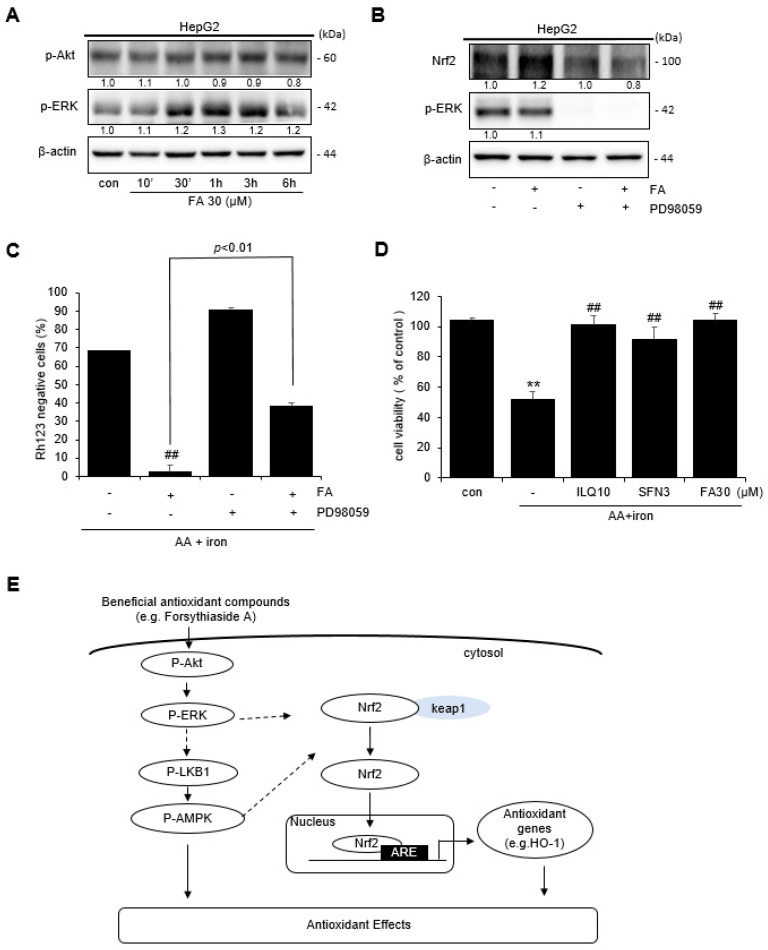
Activation of p-ERK by FA. (**A**) Western blot analysis of p-Akt and p-ERK phosphorylation. HepG2 cells were incubated with FA (30 μM) for 10 min to 6 h. (**B**) Role of p-ERK in the phosphorylation of Nrf2 by FA (30 μM). HepG2 cells were pretreated with PD98059 (30 μM) for 1 h and thereafter incubated with FA (30 μM) for 30 min. (**C**) Role of ERK in mitochondria positive effect by FA (30 μM). (**D**) MTT assay of the positive control reagents. The data represent the means ± S.D. of replicate experiments at least three times. The significance of the statistical differences between each treatment group and the vehicle-treated control group (** *p* < 0.01) or AA + iron-treated group (## *p* < 0.01) was verified. (**E**) Scheme showing the mechanism underlying the antioxidant effect of FA.

**Figure 6 ijms-24-17033-f006:**
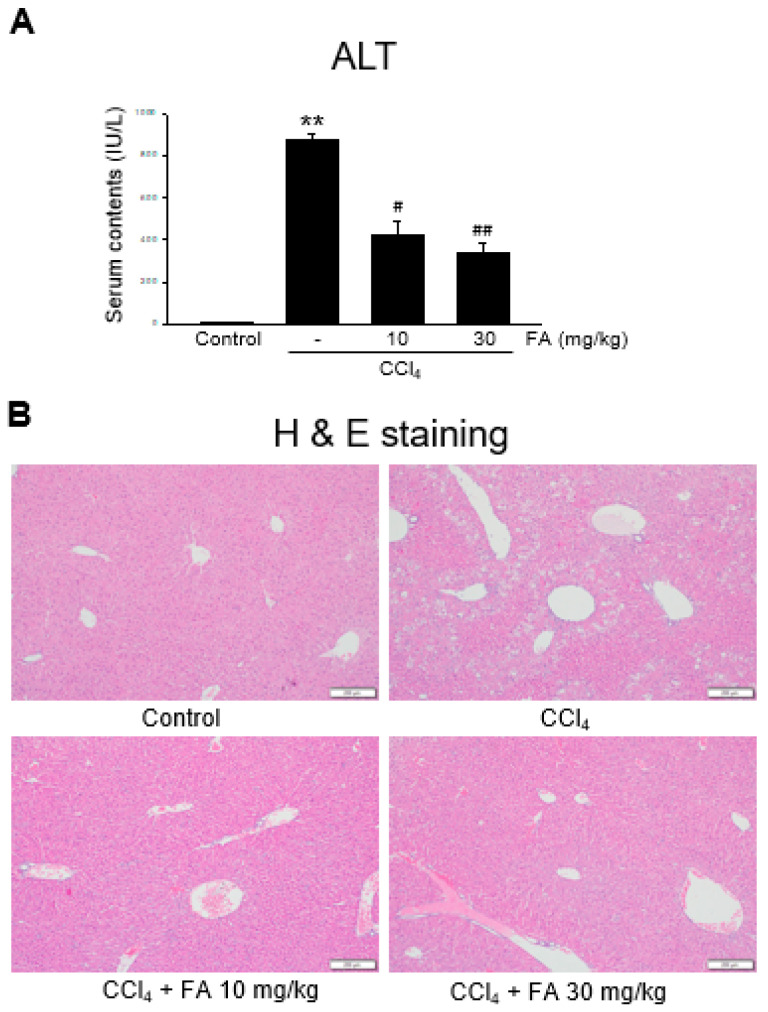
Effect of FA on CCl_4_-induced live damage in mice. (**A**) Plasma ALT levels. FA (10 and 30 mg/kg doses) were orally administrated to mice, and then CCl_4_ was injected. The data represent the mean ± S.E.M. (** *p* < 0.01 between the vehicle control; ## *p* < 0.01, # *p* < 0.05 between CCl_4_ treatments). (**B**) Histochemical analysis (×200). bar = 200 μm.

## Data Availability

The data presented in this study are available on request from the corresponding author.

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
