# Peer review of "Forsythiaside A Activates AMP-Activated Protein Kinase and Regulates Oxidative Stress via Nrf2 Signaling"

_ijms, 2023, doi:10.3390/ijms242317033_

Round 1

Reviewer 1 Report

Comments and Suggestions for Authors

The present study aims to evaluate the mechanisms involved in the hepatoprotective effects of forsythiaside A (FA). The study clearly shows that FA has pro-survival effects on HepG2 cells exposed to an arachidonic plus iron challenge. However, the further assessment of the mechanism involved falls short of its aims.

Also all the experiments should be quantified and statistically analyzed, and the text should be edited by a professional English editor.

The authors first test the cell viability of HepG2 cells following the challenge, measure cell survival but the evaluation of the possible involvement of apoptosis is insufficient (levels of pro-caspase 3 and PARP), at least caspase 3 levels and activity and the presence/absence of apoptotic nuclei should be evaluated.

The changes in ROS levels in dying cells should not be interpreted as indicative of the possible involvement of ROS in the mechanism and the presence of Rh 123 negative cells is likely more indicative of loss of mitochondrial membrane potential, linked to cell death.

The authors then test the effect of FA by itself on HepG2 cells. They show an increase in p-AMPK, but do not show the total levels of AMPK. This phosphorylation may be indicative of metabolic stress, this effect along with the activation of Nrf2 may suggest the inhibition of mitochondrial OXPHOS and the increase of mitochondrial ROS production that should, therefore, be tested.

In summary the presented results suggest a pre-conditioning mechanism for FA action both in vivo and in vitro that should be adequately tested.

Comments on the Quality of English Language

The English is good enough to understand the text but there are many obvious use of English errors that should be fixed.

Author Response

Reviewer #1

The present study aims to evaluate the mechanisms involved in the hepatoprotective effects of forsythiaside A (FA). The study clearly shows that FA has pro-survival effects on HepG2 cells exposed to an arachidonic plus iron challenge. However, the further assessment of the mechanism involved falls short of its aims.

Also all the experiments should be quantified and statistically analyzed, and the text should be edited by a professional English editor.

Answer: We changed and edited the MS the reviewer’s recommendations.

The authors first test the cell viability of HepG2 cells following the challenge, measure cell survival but the evaluation of the possible involvement of apoptosis is insufficient (levels of pro-caspase 3 and PARP), at least caspase 3 levels and activity and the presence/absence of apoptotic nuclei should be evaluated.

Answer: Thank you for pointing out the mentioned contents. To assess apoptosis, HepG2 cells were stained with annexin V/PI, and the expression of cleaved caspase 3 was examined.

The changes in ROS levels in dying cells should not be interpreted as indicative of the possible involvement of ROS in the mechanism and the presence of Rh 123 negative cells is likely more indicative of loss of mitochondrial membrane potential, linked to cell death.

Answer: Additionally, to validate the increase in mitochondrial ROS production, HepG2 cells were stained with JC-1 to measure mitochondrial membrane potential,

The authors then test the effect of FA by itself on HepG2 cells. They show an increase in p-AMPK, but do not show the total levels of AMPK. This phosphorylation may be indicative of metabolic stress, this effect along with the activation of Nrf2 may suggest the inhibition of mitochondrial OXPHOS and the increase of mitochondrial ROS production that should, therefore, be tested.

In summary the presented results suggest a pre-conditioning mechanism for FA action both in vivo and in vitro that should be adequately tested.

Answer: As a reviewer’s recommendations, MitoSOX Red was also used to stain for mitochondrial reactive oxygen species, particularly superoxide in the new MS.

Reviewer 2 Report

Comments and Suggestions for Authors

The article by Young Eun Kim et al. deals with the protective effects of Forsythiaside A during oxidative stress induced by arachidonic acid plus iron in vitro model and carbon tetrachloride (CClâ‚„)-induced mouse liver damage in vivo.

The results of the article suggested that the antioxidant and hepatoprotective effects of Forsythiaside A, an isolated active constituent from Forsythia suspensea, are due to induction of HO-1 expression by activating Nrf2 via the ERK-AMPK signaling pathway.

My comments:

p. 2.

- The authors wrote

However, in the FA-treated cells, the cell viability decreased as the concentration of AA + iron increased (Fig 1A).

But according to the legend of Fig. 1A, the concentration of AA + Iron is constant.

p. 3.

- Should be … with 30 μM FA… instead of … with 30 M FA…

- I could not find the explanation of the ACC, ECH, HO-1, etc abbreviations.

- Please, explain the method of evaluation of mitochondria dysfunction using dye Rh 123.

p. 5.

- It is difficult to confirm the increase in the level of HO-1 from the Fig. 4D.

p. 8.

- Please, explain the phrase

apoptosis of steroid hormones and calcium ions

Comments on the Quality of English Language

The article by Young Eun Kim et al. deals with the protective effects of Forsythiaside A during oxidative stress induced by arachidonic acid plus iron in vitro model and carbon tetrachloride (CClâ‚„)-induced mouse liver damage in vivo.

The results of the article suggested that the antioxidant and hepatoprotective effects of Forsythiaside A, an isolated active constituent from Forsythia suspensea, are due to induction of HO-1 expression by activating Nrf2 via the ERK-AMPK signaling pathway.

My comments:

p. 2.

- The authors wrote

However, in the FA-treated cells, the cell viability decreased as the concentration of AA + iron increased (Fig 1A).

But according to the legend of Fig. 1A, the concentration of AA + Iron is constant.

p. 3.

- Should be … with 30 μM FA… instead of … with 30 M FA…

- I could not find the explanation of the ACC, ECH, HO-1, etc abbreviations.

- Please, explain the method of evaluation of mitochondria dysfunction using dye Rh 123.

p. 5.

- It is difficult to confirm the increase in the level of HO-1 from the Fig. 4D.

p. 8.

- Please, explain the phrase

apoptosis of steroid hormones and calcium ions

Author Response

Reviewer #2

Comments and Suggestions for Authors

The article by Young Eun Kim et al. deals with the protective effects of Forsythiaside A during oxidative stress induced by arachidonic acid plus iron in vitro model and carbon tetrachloride (CClâ‚„)-induced mouse liver damage in vivo.

The results of the article suggested that the antioxidant and hepatoprotective effects of Forsythiaside A, an isolated active constituent from Forsythia suspensea, are due to induction of HO-1 expression by activating Nrf2 via the ERK-AMPK signaling pathway.

My comments:

  1. 2.

- The authors wrote

However, in the FA-treated cells, the cell viability decreased as the concentration of AA + iron increased (Fig 1A).

But according to the legend of Fig. 1A, the concentration of AA + Iron is constant.

-> Thank you for pointing that out. I revised the part mentioned

  1. 3.

- Should be … with 30 μM FA… instead of … with 30 M FA…

- I could not find the explanation of the ACC, ECH, HO-1, etc abbreviations.

- Please, explain the method of evaluation of mitochondria dysfunction using dye Rh 123.

-> Thank you for commenting on this. I modified it to 30 μM FA, and added a description of the mitochondrial dysfunction evaluation method using dye Rh123 and abbreviations such as ACC, ECH, and HO-1.

  1. 5.

- It is difficult to confirm the increase in the level of HO-1 from the Fig. 4D.

-> Thank you for talking about this. It was replaced by mRNA expression analysis by Real-Time PCR.

  1. 8.

- Please, explain the phrase

apoptosis of steroid hormones and calcium ions

-> Upon reviewing the points mentioned by the reviewer, I realized that there were inaccuracies in the content, so I have made corrections.

Comments on the Quality of English Language

The article by Young Eun Kim et al. deals with the protective effects of Forsythiaside A during oxidative stress induced by arachidonic acid plus iron in vitro model and carbon tetrachloride (CClâ‚„)-induced mouse liver damage in vivo.

Reviewer 3 Report

Comments and Suggestions for Authors

The manuscript “Forsythiaside A activates AMP-activated protein kinase and regulates the oxidative stress in the liver via Nrf2 signaling” by Young Eun Kim et al.  The authors have performed various experiments to prove their hypothesis. However, I have several comments that need to be addressed.

1.     Abstract: I will suggest rewriting the conclusion section in a better way.

2.     I will suggest to elaborate introduction section.

3.     How animals were sacrificed…anesthesia was used?...must be described in the manuscript.

4.     In methodology western blot is missing…. must be described.

5.     How much total proteins were loaded for western blot, must be incorporated in the manuscript.

6.     I will suggest adding, primary and secondary antibodies dilution in the manuscript.

7.     I will suggest adding the specific molecular weights of all the proteins in western blot images.

8.     In histology images, I will suggest adding magnification.

9.     I will suggest that authors can add a few gene expression studies in this manuscript that can strengthen the data presented.

10.  The abbreviation throughout the manuscript should be checked.

11.  Minor grammatical errors should be corrected.

Author Response

Reviewer #3

The manuscript “Forsythiaside A activates AMP-activated protein kinase and regulates the oxidative stress in the liver via Nrf2 signaling” by Young Eun Kim et al.  The authors have performed various experiments to prove their hypothesis. However, I have several comments that need to be addressed.

  1. Abstract: I will suggest rewriting the conclusion section in a better way.

-> Thank you for a good idea. I revised the conclusion according to the reviewer's opinion.

  1. I will suggest to elaborate introduction section.

-> Thank you for your recommendations. We revised the introductions according to the reviewer's opinion.

  1. How animals were sacrificed…anesthesia was used?...must be described in the manuscript.

-> Thank you for pointing that out. I explained to the manuscript about animal sacrifice and use of anesthesia.

  1. In methodology western blot is missing…. must be described.

-> I added a description of the western blot to the reviewer's opinion.

  1. How much total proteins were loaded for western blot, must be incorporated in the manuscript.

-> Thank you for commenting on this. According to the reviewer's opinion and suggestion, the contents of the total proteins amount have been revised.

  1. I will suggest adding, primary and secondary antibodies dilution in the manuscript.

-> As suggested by the reviewer, I added primary antibody dilution and secondary antibody dilution in the manuscript.

  1. I will suggest adding the specific molecular weights of all the proteins in western blot images.

-> Thank you for talking about this. I added a specific molecular weight of all the proteins in the western blot image.

  1. In histology images, I will suggest adding magnification.

-> I added magnification to the historical image.

  1. I will suggest that authors can add a few gene expression studies in this manuscript that can strengthen the data presented.

-> According to the reviewer's opinion, mRNA expression analysis by Real-Time PCR was added.

  1. The abbreviation throughout the manuscript should be checked.

-> Thank you for pointing that out. I checked and corrected the abbreviations in the manuscript.

  1. Minor grammatical errors should be corrected.

-> According to the reviewer’s comment, I checked and corrected the grammatical errors in the manuscript.

Round 2

Reviewer 1 Report

Comments and Suggestions for Authors

The authors hava addressed some of the previously indicated issues but there are some that still need to be addressed.

The most important relates to the evaluation of the mechanism involved. As indicated before, the authors should test the response of HepG2+FA in the absence of iron, evaluating mitochondrial respiration (ie SeaHorse, Oroboros, Clark electrode) and mitochondrial ROS (ie MitoSOX), since the data presented, in the presence of iron, and with cells undergoing apoptosis, is not enough to conclude that the protection mechanism relates to mitochondrial ROS production

Also, not just the phosphorylated form but the total levels  of AMPK, AKT and ACC should be tested.

And, last but not least, ALL the blots and images should be quantified and graphed.

Comments on the Quality of English Language

The English is gramatically correct by the style, the "use of English" could be improved.

Author Response

Reviewer #1

The authors hava addressed some of the previously indicated issues but there are some that still need to be addressed.

   Answer: We appreciate the reviewer's valuable comments.

The most important relates to the evaluation of the mechanism involved. As indicated before, the authors should test the response of HepG2+FA in the absence of iron, evaluating mitochondrial respiration (ie SeaHorse, Oroboros, Clark electrode) and mitochondrial ROS (ie MitoSOX), since the data presented, in the presence of iron, and with cells undergoing apoptosis, is not enough to conclude that the protection mechanism relates to mitochondrial ROS production

   Answer: Thank you once again for your feedback. In response to the reviewer's comments, we assessed the effects of FA on HepG2 cells in the absence of iron. To measure mitochondrial membrane potential, JC-1 staining was performed on HepG2 cells. Additionally, to measure mitochondrial ROS, HepG2 cells were stained with MitoSOX Red and analyzed.

Also, not just the phosphorylated form but the total levels of AMPK, AKT and ACC should be tested.

   Answer: Thank you for pointing that out. We verified the total levels of the proteins. The experimental results have been added to Figure 3A.

And, last but not least, ALL the blots and images should be quantified and graphed.

   Answer: Thank you for talking about this. We added the quantification results and graphs to the new manuscript.

Reviewer 3 Report

Comments and Suggestions for Authors

Authors have responded all the comments

Author Response

Authors have responded all the comments

Answer: We appreciate your recommendations. Thank you.